# Effects of High-Flow Nasal Cannula on Right Heart Dysfunction in Patients with Acute-on-Chronic Respiratory Failure and Pulmonary Hypertension

**DOI:** 10.3390/jcm12175472

**Published:** 2023-08-23

**Authors:** Corrado Pelaia, Giuseppe Armentaro, Chiara Lupia, Antonio Maiorano, Nicola Montenegro, Sofia Miceli, Valentino Condoleo, Velia Cassano, Andrea Bruni, Eugenio Garofalo, Claudia Crimi, Alessandro Vatrella, Girolamo Pelaia, Federico Longhini, Angela Sciacqua

**Affiliations:** 1Department of Health Sciences, University “Magna Græcia” of Catanzaro, 88100 Catanzaro, Italy; pelaia.corrado@gmail.com (C.P.); chiaralupia1996@gmail.com (C.L.); antoniomaiorano95@gmail.com (A.M.); n.montenegro@hotmail.it (N.M.); pelaia@unicz.it (G.P.); 2Department of Medical and Surgical Sciences, University “Magna Græcia” of Catanzaro, 88100 Catanzaro, Italy; giuseppearmentaro91@gmail.com (G.A.); sofy.miceli@libero.it (S.M.); condoleovalentino@gmail.com (V.C.); velia.cassano@libero.it (V.C.); andreabruni87@gmail.com (A.B.); eugenio.garofalo@gmail.com (E.G.); sciacqua@unicz.it (A.S.); 3Department of Clinical and Experimental Medicine, University of Catania, 95123 Catania, Italy; dott.claudiacrimi@gmail.com; 4Department of Medicine, Surgery and Dentistry, University of Salerno, 84081 Salerno, Italy; avatrella@unisa.it

**Keywords:** high-flow nasal cannula, high-flow nasal therapy, conventional oxygen therapy, acute-on-chronic respiratory failure, pulmonary hypertension, echocardiography, hemodynamic, right ventricle

## Abstract

High-flow nasal cannula (HFNC) has several benefits in patients affected by different forms of acute respiratory failure, based on its own mechanisms. We postulated that HFNC may have some advantages over conventional oxygen therapy (COT) on the heart function in patients with acute-on-chronic respiratory failure with concomitant pulmonary hypertension (PH). We therefore designed this retrospective observational study to assess if HFNC improves the right and left ventricle functions and morphologies, arterial blood gases (ABGs), and patients’ dyspnea, compared to COT. We enrolled 17 hospitalized patients receiving HFNC, matched with 17 patients receiving COT. Echocardiographic evaluation was performed at the time of admission (baseline) and 10 days after (T_10_). HFNC showed significant improvements in right ventricular morphology and function, and a reduction in sPAP. However, there were no significant changes in the left heart measurements with HFNC application. Conversely, COT did not lead to any modifications in echocardiographic measurements. In both groups, oxygenation significantly improved from baseline to T_10_ (in the HFNC group, from 155 ± 47 to 204 ± 61 mmHg while in the COT group, from 157 ± 27 to 207 ± 27 mmHg; *p* < 0.0001 for both comparisons). In conclusion, these data suggest an improvement of oxygenation with both treatments; however, only HFNC was able to improve the right ventricular morphology and function after 10 days from the beginning of treatment in a small cohort of patients with acute-on-chronic respiratory failure with PH.

## 1. Introduction

Pulmonary hypertension (PH) is a condition characterized by a mean pulmonary arterial pressure (mPAP) greater than 20 mmHg [1]. The primary form is represented by group 3, which includes PH associated with lung diseases and accounts for approximately 90% of all PH cases [2]. The key pathogenic mechanisms that lead to PH due to pulmonary disease are represented by hypoxia-induced vasoconstriction [3], remodeling, inflammation, and thrombosis [4,5], while the main clinical symptom is dyspnea, associated with progressive exercise intolerance [6].

Treatment of PH includes calcium channel blockers, prostacyclin analogues, endothelin 1 (ET-1) receptor antagonists, phosphodiesterase-5 inhibitors, and supportive therapy with diuretics, oral anticoagulants, and especially in group 3 PH, conventional oxygen therapy (COT) [7]. Several previous studies demonstrated that oxygen supplementation decreases mPAP and improves right ventricle (RV) function [3,8,9,10].

High-flow nasal cannula (HFNC) is an innovative device that combines an air–oxygen blender with an active humidifier, capable of delivering humidified gas at a flow rate of up to 60 L/min providing a constant and precise fraction of inspired oxygen (FiO_2_), which can be set between 21% and 100% [11].

HFNC has shown several physiological benefits [12], including a reduction in work of breathing and respiratory rate, and improvement of dyspnea and respiratory dynamics, facilitated by the washout of the anatomic dead space [13]. HFNC also improves mucociliary clearance [14] and generates a modest positive end-expiratory pressure (PEEP) effect, resulting in an increase in end-expiratory lung volume (EELV) and prevention of alveolar closure [13]. Moreover, it provides good patient tolerance and comfort [15]. As a result, it could be an alternative oxygen-delivery strategy for patients with PH, theoretically combining adequate oxygenation, dyspnea relief, and patient comfort [12].

Indeed, HFNC is recommended as the first-line treatment in patients with hypoxemic respiratory failure and in selected patients with hypercapnic respiratory failure [16]. However, to date, there are no studies specifically investigating the role of HFNC in patients with PH [17].

We aim to assess if HFNC improves the right and left ventricle functions and morphologies in patients with acute-on-chronic respiratory failure and PH. Secondarily, we will assess if HFNC improves arterial blood gases (ABGs) and patients’ dyspnea compared to COT.

## 2. Materials and Methods

This observational, retrospective, case–control study was conducted at the University Hospital “Renato Dulbecco” in Catanzaro, Italy. The study adhered to the principles outlined by the Declaration of Helsinki and its amendments. Approval for the study was obtained from the local Ethics Committee “Comitato Etico Sezione Area Centro—Regione Calabria” under protocol number 118/2023 on 20 April 2023. Written informed consent was obtained from all participating patients. Additionally, all individual datasets generated during and/or analyzed throughout the study have been de-identified and are available from the corresponding author upon reasonable request.

### 2.1. Patients

We included all adult patients (i.e., ≥18 years old) who were referred to the Respiratory Unit with a diagnosis of acute-on-chronic respiratory failure and PH. The diagnosis of acute-on-chronic respiratory failure was defined as worsening dyspnea associated with an arterial partial pressure of oxygen (PaO_2_) < 60 mmHg on room air [18].

Patients with any of the following conditions were excluded from the study: (1) inability to provide informed consent; (2) withdrawal of consent; (3) active cancer; (4) advanced liver cirrhosis (Child–Pugh C); (5) renal insufficiency (baseline estimated glomerular filtration rate—eGFR < 30 mL/min/1.73 m^2^); and (6) inclusion in other ongoing research protocols.

### 2.2. High Flow through Nasal Cannula (HFNC) and Conventional Oxygen Therapy (COT)

The group of patients who received HFNC was treated with a dedicated device (AIRVO2, Fisher&Paykel Healthcare, Auckland, New Zealand). The size of the nasal cannula was chosen to occlude approximately 2/3 of the patient’s nostril. The flow rate and temperature were initially set at 60 L/min and 37 °C, respectively. However, in case of discomfort, the flow and/or temperature were adjusted to the most tolerated setting [13,19]. The FiO_2_ was set to maintain a peripheral oxygen saturation (SpO_2_) level above 90%.

On the other hand, COT was administered through a Venturi mask. Like the HFNC group, the FiO_2_ was adjusted to maintain SpO_2_ above 90%. Patients in the COT group were matched for age, sex, BMI, PH etiology, and therapy.

### 2.3. Data Collection and Echocardiographic Assessments

All data were collected and recorded in a custom-made database using Microsoft Excel (Microsoft, Washington, DC, USA). Anthropometric data, including age, gender, body mass index, and smoking habits, were recorded, along with information on pharmacological therapy (use of antifibrotic therapy, inhaled treatment, antibiotics, and diuretics), and the presence of comorbidities at hospital admission (baseline). Additionally, data regarding TransThoracic Echocardiography (TTE) and arterial blood gases (ABGs) were collected at baseline and around 10 days after the initiation of patients’ treatment (T_10_).

TTE was performed following the recommendations of the American Society of Echocardiography (ASE) [20]. Recordings were conducted using a VIVID 7 Pro ultrasound system (GE Technologies, Milwaukee, WI, USA) and a 2.5 MHz transducer. All assessments were performed by a single operator who was an expert in the technique.

Tricuspid regurgitant velocity (TRV) was analyzed by continuous Doppler at the level of the atrioventricular plane of the tricuspid valve, either in projection with the four apical chambers or, in the case of eccentric jets, in parasternal short axis. Subsequently, the systolic pulmonary arterial pressure (sPAP) was derived using the Bernoulli equation to assess the right ventricular (RV) systolic pressure [20]. Measurements of the right ventricular outflow tract (RVOT) diameter, right atrium area (RAA), inferior vena cava (IVC) diameter, interventricular septal thickness (IVST), left ventricular ejection fraction (LVEF), and left atrial volume index (LAVI) were obtained according to ASE recommendations [20]. The tricuspid annular plane systolic excursion (TAPSE), which reflects the right longitudinal function, was recorded at the free wall of the RV by assessing the movement of the tricuspid annulus [21]. Furthermore, we calculated the TAPSE/sPAP ratio to assess the RV length/strength relationship [22,23].

ABG measurements included PaO_2_, partial arterial pressure of carbon dioxide (PaCO_2_), pH, and bicarbonates (HCO_3_^−^).

Patients’ dyspnea was measured using an 11-point Numeric Rating Scale. After providing a detailed explanation before initiating the protocol, patients were asked to indicate a number between 0 (no discomfort/dyspnea) and 10 (worst possible discomfort/dyspnea) on an adapted printed scale [24,25].

### 2.4. Statistical Analysis

To assess the normal distribution of data, we utilized the Anderson–Darling test. Continuous variables were presented as either mean ± standard deviation (SD) or median [25th–75th interquartile range] (IQR) based on their normal distribution. Categorical data were expressed as counts and percentages. For comparisons between groups, we employed the paired *t*-test or Mann–Whitney U-test for continuous variables, and Fisher’s test for categorical variables.

In both the HFNC and COT cohorts of patients, we computed the differences from T_10_ to baseline for all recorded parameters and compared them between treatments. Mean differences with corresponding 95% confidence intervals were calculated to evaluate the treatment effects.

To assess relationships between variables, we used the Pearson correlation coefficient (r) and performed linear regression analysis.

Statistical significance was set at a *p*-value < 0.05 for all tests. All statistical analyses were conducted using Prism version 9.4.0 (GraphPad Software Inc., San Diego, CA, USA).

## 3. Results

We collected data from 34 adult patients, with 17 receiving HFNC and 17 receiving COT, from September 2019 to September 2020. The enrollment flowchart is reported in Figure 1. In particular, 436 patients were hospitalized in the study period with acute-on-chronic respiratory failure with PH. TTE was available at baseline and after 10 days in 91 patients; among these patients, 17 received HFNC during their hospital stay, and they have been matched with another 17 patients receiving COT.

Patients’ characteristics are shown in Table 1. The two study groups were balanced according to gender, body mass index, comorbidities, and therapy. The mean length of stay was similar between the HFNC and COT groups (11.8 ± 4.08 vs. 12.1 ± 4.5 days; *p* = 0.999).

### 3.1. TTE Measurements

Data from TTE are presented in Table 2. HFNC demonstrated significant improvements in right ventricular morphology and function, along with a reduction in sPAP. However, no significant changes were observed in the left heart measurements following the application of HFNC. On the other hand, COT did not lead to any modifications in TTE measurements.

Table 3 displays the treatment effect sizes of HFNC compared to COT on acquired TTE assessments at T_10_. HFNC exhibited its most substantial effect in reducing sPAP, improving right ventricle contraction (measured by TAPSE and TAPSE/sPAP), and slightly reducing preload (measured by IVC and RAA) when compared to COT.

### 3.2. Arterial Blood Gases and Comfort

In the HFNC cohort, oxygenation (i.e., PaO_2_/FiO_2_) significantly increased from 155 ± 47 to 204 ± 61 (*p* < 0.001). On the contrary, pH (7.42 ± 0.05 vs. 7.42 ± 0.03; *p* = 0.665), PaCO_2_ (49.2 ± 16.2 vs. 46.4 ± 11.1 mmHg; *p* = 0.234), and HCO_3_^−^ (30.4 ± 6.8 mmol/L vs. 30.0 ± 6.2 mmol/L; *p* = 0.627) did not change from baseline to T_10_.

In the COT group, we also recorded a significant improvement of PaO_2_/FiO_2_ from 157 ± 27 to 207 ± 27 (*p* < 0.0001). In addition, no significant modifications were detected with respect to pH (7.41 ± 0.04 vs. 7.41 ± 0.03; *p* = 0.919), PaCO_2_ (45.0 ± 10.0 mmHg vs. 45.6 ± 7.4 mmHg; *p* = 0.734), and HCO_3_^−^ (28.8 ± 5.8 mmol/L vs. 28.6 ± 4.0 mmol/L; *p* = 0.898).

Dyspnea was reported as reduced from 7 [7; 8] to 4 [4; 5] in the HFNC group and from 7 [7; 8] to 5 [4; 6] in the COT group (*p* < 0.001 for both comparisons).

At T_10_, we did not find any effect of the HFNC over COT on PaO_2_/FiO_2_ (−1.42, 95% CI [−26.33; 23.50]; *p* = 0.908), pH (−0.01, 95% CI [−0.03; 0.02]; *p* = 0.693), PaCO_2_ (−3.35, 95% CI [−9.19; 2.49]; *p* = 0.251), HCO_3_^−^ (−0.26, 95% CI [−3.95; 3.43]; *p* = 0.886), and VAS (−0.64, 95% CI [−1.37; 0.08]; *p* = 0.081).

### 3.3. Correlation between TTE Measurements and Oxygenation Changes

In the HFNC group, only an sPAP decrease was found to be inversely correlated with PaO_2_/FiO_2_ (r = −0.512; *p* < 0.05). Conversely, no other statistically significant correlations were detected between PaO_2_/FiO_2_ change and variations in RAA (r = −0.09; *p* = 0.711), IVC diameter (r = −0.265; *p* = 0.302), TAPSE (r = 0.204; *p* = 0.433), TAPSE/sPAP (r = 0.273; *p* = 0.289), IVST (r = 0.162; *p* = 0.701), RVOT (r = 0.264; *p* = 0.494), LAVI (r = 0.089; *p* = 0.819), LVEF (r = 0.294; *p* = 0.288), and aortic root dimensions (r = −0.031; *p* = 0.936).

In the COT cohort of patients, no correlations were found between PaO_2_/FiO_2_ changes and variations in RAA (r = 0.128; *p* = 0.724), IVC diameter (r = −0.063; *p* = 0.854), sPAP (r = −0.235; *p* = 0.441), TAPSE (r = 0.163; *p* = 0.563), TAPSE/sPAP (r = 0.613; *p* = 0.106), IVST (r = −0.276; *p* = 0.439), RVOT (r = 0.467; *p* = 0.198), LAVI (r = 0.134; *p* = 0.731), LVEF (r = 0.140; *p* = 0.647), and aortic root dimensions (r = 0.114; *p* = 0.739).

## 4. Discussion

This study demonstrates that, in patients with acute-on-chronic respiratory failure and concomitant PH, both HFNC and COT improve oxygenation at T_10_, although only HFNC reduces sPAP and right ventricle function.

Several studies have reported that HFNC therapy improves oxygenation and reduces dyspnea in patients with hypoxemia of different origins and severity. However, few studies have explored the hemodynamic effects of HFNC and how the cardiopulmonary interactions may be influenced by the small changes in intrathoracic pressure caused by the flow delivered, as well as by the positive end-expiratory pressure effects generated, particularly in decompensated chronic pulmonary hypertension.

We believe that the observed reduction in dyspnea and hemodynamic effects are mainly due to an improvement in the ventilation-perfusion ratio and lung diffusion capacity (due to the small applied positive expiratory pressure), improvement in gas exchange (by the administration of more stable FiO_2_), and the administration of heated and humidified air –oxygen admixture. Indeed, chronic lung diseases are characterized by abnormalities of both the ventilation–perfusion ratio and lung diffusion capacity, resulting in gas exchange impairment [26]. Though correction of hypoxemia is one of the cornerstones of PH treatment [26], the use of HFNC may be controversial. HFNC generates a PEEP up to 7.4 cm H_2_O [27]. In patients with severe PH, this effect may increase pulmonary vascular resistance (PVR) and pulmonary artery pressures, exacerbating the hemodynamic burden on the right ventricle. In such cases, caution is warranted, and careful monitoring of the patient’s response is essential. Conversely, in patients with less severe PH, HFNC therapy may have a modest beneficial effect on pulmonary artery pressures. Of note, PEEP generated by HFNC maintains alveolar recruitment with a favorable effect on pulmonary artery pressures [13]. Beside the alveoli recruitment or prevention of further alveolar collapse [13,28,29], these effects are involved in lowering the right ventricular preload and the left ventricular afterload in patients with heart failure and acute cardiogenic pulmonary edema [30,31,32,33].

It should also be remarked that HFNC delivers a warm and humidified air–oxygen admixture, with a stable and precise FiO_2_ [13,25], which can mitigate hypoxemia [25] and reduce the strain on the pulmonary circulation. HFNC decreases the anatomic dead space by washing out carbon dioxide from the upper airways [34], and consequently may improve dyspnea while preserving comfort for the patient [35].

In this study, most enrolled patients had underlying interstitial lung diseases (ILD), which typically have a restrictive impairment characterized by a decrease in forced vital capacity, total lung capacity, and diffusing lung capacity for carbon monoxide, as well as a decreased exercise capacity. The reduction in lung compliance causes an increase in respiratory rate as a compensatory mechanism, while hypercapnia occurs only in the late phase of such diseases [36]. Within this context, hypoxemia-dependent pulmonary vasoconstriction leads to an increase in pulmonary arterial pressure [37]. Because of disease worsening, there will be a reduction in the vascular bed, an increase in the right ventricular afterload, and, ultimately, heart failure [38]. The clinical application of HFNC in ILDs with chronic respiratory failure requires further elucidation. However, compared to COT, HFNC may have a role in these patients as a palliative treatment [39].

We wondered if there were some beneficial effects of HFNC on dysfunctional right heart in patients with acute-on-chronic respiratory failure and PH. The available evidence is still not conclusive. In a study of 10 patients with heart failure and New York Heart Association (NYHA) class III, Roca et al. demonstrated that after a 60 min trial with HFNC, there was a reduction in the inspiratory collapse of the IVC and right ventricular preload, while sPAP did not change [40]. In our study, we assessed the hemodynamic effects over a much longer time interval, demonstrating a beneficial effect on a wider range of transesophageal echocardiography (TEE) parameters, including IVC diameter, right ventricle function, and sPAP. Similar findings were reported by Gupta et al. when studying three elderly and postpartum patients with PH, type I respiratory failure, and associated comorbidities [41]. The first two patients manifested PH as a consequence of valvular heart disease and pulmonary tuberculosis, respectively. The third one had primary PH [41]. However, they experienced only decreases in respiratory rate, improvements in oxygen saturation and PaO_2_, as well as subjective comfort using the HFNC device. Moreover, these patients did not undergo instrumental exams that could specifically prove an improvement in the right ventricle function and a reduction in PH.

To the best of our knowledge this is the first study that showed that HFNC therapy in patients with chronic lung diseases and concomitant PH can induce a greater improvement of right ventricle preload and function, compared to COT.

Our study has limitations. First, the small sample size and the single-center design may limit the generalizability of our findings. Second, we performed TEE but not a lung volume or airway pressure assessment [42]. As a result, based on the current literature, we can only speculate that HFNC improves PH and right heart performance because of a slight increase in intrathoracic pressure. Third, it would be interesting to assess if HFNC may improve some clinical outcomes as opposed to COT; in addition, it would be also relevant to compare patients with interstitial lung disease separately from those with chronic obstructive pulmonary disease. However, the small sample of patients precludes us from the possibility to perform such an analysis. Future research with a larger sample size should be focused on evaluating the effects of HFNC on clinical outcome and in patients with chronic lung diseases separately from those with interstitial lung disease with concomitant PH.

## 5. Conclusions

In conclusion, HFNC ameliorates the right ventricle preload and function and sPAP in patients with acute-on-chronic respiratory failure and concomitant PH.

Although these are promising data, we cannot provide any definitive conclusion; further studies are therefore required to better understand the underlying mechanisms in this population of patients.

## Figures and Tables

**Figure 1 jcm-12-05472-f001:**
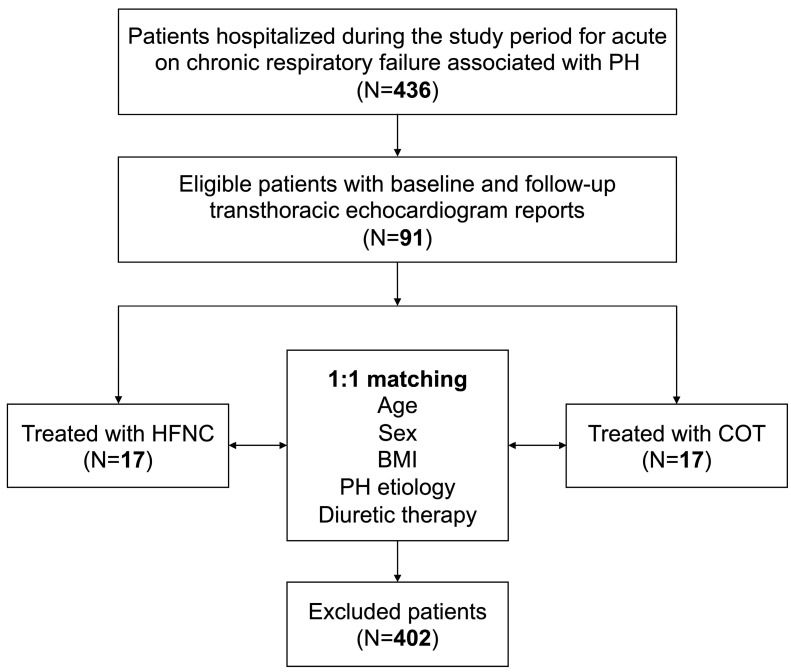
Study flow diagram. PH, pulmonary hypertension; HFNC, high-flow nasal cannula; COT, conventional oxygen therapy; and BMI, body mass index.

**Table 1 jcm-12-05472-t001:** Baseline patient characteristics.

	HFNC (*n* = 17)	Control (*n* = 17)	*p*
Age, years, mean, (SD)	67.82 (7.74)	71.06 (6.18)	0.187
Male gender, *n* (%)	12 (70.59)	12 (70.59)	>0.999
BMI, mean (SD), kg/m^2^	25.76 (4.19)	25.06 (4.31)	0.632
Smoking history, *n* (%)	13 (76.47)	11 (64.71)	0.708
Pulmonary hypertension etiology			
ILD, *n* (%)	7 (41.18)	7 (41.18)	>0.999
OSAS, *n* (%)	2 (11.76)	2 (11.76)	>0.999
COPD, *n* (%)	6 (35.29)	6 (35.29)	>0.999
Chronic pulmonary embolism, *n* (%)	2 (11.76)	2 (11.76)	>0.999
Depression, *n* (%)	3 (17.65)	4 (23.53)	>0.999
Antibiotic therapy, *n* (%)	14 (82.35)	12 (70.59)	0.688
Diuretic therapy, *n* (%)	13 (76.47)	13 (76.47)	>0.999
Antifibrotic therapy, *n* (%)	6 (35.29)	6 (35.29)	>0.999
Inhaled therapy, *n* (%)	6 (35.29)	7 (41.18)	>0.999
LAMA, *n* (%)	0 (0.00)	1 (5.88)	>0.999
LABA + LAMA, *n* (%)	2 (11.76)	1 (5.88)	>0.999
ICS + LABA + LAMA, *n* (%)	4 (23.53)	5 (29.41)	>0.999

Abbreviations: SD, standard deviation; HFNC, high-flow nasal cannula; BMI, body mass index; ILD, interstitial lung disease; OSAS, obstructive sleep apnea syndrome; COPD, chronic obstructive pulmonary disease; GERD, gastroesophageal reflux disease; LAMA, long-acting muscarinic receptor antagonist; LABA, long-acting β_2_-adrenergic agonist; and ICS, inhaled corticosteroid.

**Table 2 jcm-12-05472-t002:** Cardiac ultrasonography assessments in HFNC and COT cohorts.

	HFNC			COT		
	Baseline	T_10_	*p*-Value Baseline vs. T_10_	Baseline	T_10_	*p*-Value Baseline vs. T_10_
Right Heart						
IVC (mm)	24 [21; 27]	21 [17; 23]	0.014	19 [19; 24]	20 [19; 24]	0.999
RAA (cm^2^)	28.1 ± 9.9	23.3 ± 7.6	0.011	28.5 ± 6.5	27.0 ± 6.6	0.742
TAPSE (mm)	19.8 ± 3.4	21.5 ± 3.3	0.036	20.0 ± 4.3	20.8 ± 4.2	0.265
IVST (cm)	1.4 ± 0.2	1.3 ± 0.1	0.685	1.3 ± 0.1	1.3 ± 0.1	0.999
RVOT (cm)	3.6 ± 0.8	3.4 ± 0.7	0.420	3.5 ± 0.7	3.5 ± 0.7	0.632
sPAP (mmHg)	68 ± 21	56 ± 21	0.001	65 ± 18	66 ± 16	0.700
TAPSE/sPAP (mm/mmHg)	0.33 ± 0.14	0.45 ± 0.22	0.001	0.32 ± 0.08	0.33 ± 0.08	0.219
Left heart						
LAVI (mL/m^2^)	36.5 [27.3; 45.8]	35.5 [27.5; 36.0]	0.125	30.5 [26.3; 40.0]	27.0 [23.3; 43.3]	0.719
LVEF (%)	55.9 ± 4.2	56.1 ± 2.3	0.813	55.2 ± 6.2	54.6 ± 5.9	0.668
Aortic root (cm)	3.6 ± 0.4	3.7 ± 0.4	0.4571	3.9 ± 0.5	3.9 ± 0.5	0.052

Abbreviations: IVC, inferior vena cava; RAA, right atrium area; TAPSE, tricuspid annular plane systolic excursion; IVST, interventricular septal thickness; RVOT, right ventricular outflow tract; sPAP, systolic pulmonary arterial pressure; TAPSE/sPAP, ratio between TAPSE and sPAP; LAVI, left atrial volume index; LVEF, left ventricular ejection fraction. Data are expressed as mean ± SD or median [25th; 75th percentile]. *p*-values < 0.05 are considered significant.

**Table 3 jcm-12-05472-t003:** Treatment effects on cardiac ultrasonography assessments at T_10._

	HFNC vs. COT	
	*p*-Value	Mean Difference [95% CI]
Right Heart		
IVC (mm)	0.043	−3.46 [−6.80; −0.12]
RAA (cm^2^)	0.033	−3.52 [−6.74; −0.30]
TAPSE (mm)	0.045	2.00 [0.05; 3.95]
IVST (cm)	0.822	−0.02 [−0.15; 0.12]
RVOT (cm)	0.857	−0.04 [−0.5; 0.38]
sPAP (mmHg)	0.002	−13.05 [−20.64; −5.25]
TAPSE/sPAP (mm/mmHg)	0.030	0.11 [0.01; 0.20]
Left heart		
LAVI (mL/m^2^)	0.495	−11.67 [−47.08; 23.75]
LVEF (%)	0.808	0.37 [−2.74; 3.48]
Aortic root (cm)	0.067	−0.35 [−1.23; 0.52]

Abbreviations: IVC, inferior vena cava; RAA, right atrium area; TAPSE, tricuspid annular plane systolic excursion; IVST, interventricular septal thickness; RVOT, right ventricular outflow tract; sPAP, systolic pulmonary arterial pressure; TAPSE/sPAP, ratio between TAPSE and sPAP; LAVI, left atrial volume index; LVEF, left ventricular ejection fraction. *p* values < 0.05 are considered significant.

## Data Availability

The authors will share all the individual participant data collected during the trial after de-identification, to researchers who provide a methodologically sound proposal. The full protocol and raw data are available at flonghini@unicz.it.

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
