# Peer review of "Effects of High-Flow Nasal Cannula on Right Heart Dysfunction in Patients with Acute-on-Chronic Respiratory Failure and Pulmonary Hypertension"

_jcm, 2023, doi:10.3390/jcm12175472_

Round 1

Reviewer 1 Report

Dear authors

This study likes very interesting

I suggest explain better how you mattched the patients in both groups; how many patients were analysed at the begin in every group?

I consider that the authors should review and change, if they consider, the following sentence "Though correction of hypoxemia is one of the cornerstones of PH treatment [32], the use of HFNC may be controversial. HFNC generates a PEEP up to 7.4 cmH2O [33,34]. In patients with severe PH, this effect may increase pulmonary vascular resistance (PVR) and pulmonary artery pressures, exacerbating the hemodynamic burden on the right ventricle."

The authors commented that "In this study, most enrolled patients had underlying Interstitial Lung Diseases" (41%) but the COPD was quite similar (35%). I consider the authors can´t conclude that HFNC has only  beneficial effect over lung disease. The simple size is small but probably a subanalysis comparing the effect of HFNC in both gropus of lung diseases (ILD vs COPD) would be interesting for this study.

I waif for your answer.

Best regards

Author Response

Dear authors, this study likes very interesting

  • I suggest explain better how you mattched the patients in both groups; how many patients were analysed at the begin in every group?

Following the Reviewer’s indication, we have better specified the criteria to match patients (materials and methods) and the patients’ study flow (in the beginning of the results section).

  • I consider that the authors should review and change, if they consider, the following sentence "Though correction of hypoxemia is one of the cornerstones of PH treatment [32], the use of HFNC may be controversial. HFNC generates a PEEP up to 7.4 cmH2O [33,34]. In patients with severe PH, this effect may increase pulmonary vascular resistance (PVR) and pulmonary artery pressures, exacerbating the hemodynamic burden on the right ventricle."

Unfortunately, it is not clear what should be modified according to the Reviewer. If he/she can indicate more specifically his/her suggestion, we can proceed following the indication.

  • The authors commented that "In this study, most enrolled patients had underlying Interstitial Lung Diseases" (41%) but the COPD was quite similar (35%). I consider the authors can´t conclude that HFNC has only beneficial effect over lung disease. The simple size is small but probably a subanalysis comparing the effect of HFNC in both gropus of lung diseases (ILD vs COPD) would be interesting for this study.

The reviewer is right and a subanalysis would be very interesting. However, given the small sample of patients, we are unable to perform a subanalysis of the two study groups. We have now discussed this issue as limitation of the study.

Reviewer 2 Report

Thank you for your submission. Overall, it is a very well structured and well presented scientific work. 

The only minor comment I could add is that it would be interesting to add a short paragraph at the discussion session and give further information about the dyspnea and comfort of the patients using HFNC over COT, as you have shown that there has been a clinically significant difference in the dyspnea scale between the two groups.

Author Response

  • Thank you for your submission. Overall, it is a very well structured and well presented scientific work. 

We thank the Reviewer for his/her appreciation of our study.

  • The only minor comment I could add is that it would be interesting to add a short paragraph at the discussion session and give further information about the dyspnea and comfort of the patients using HFNC over COT, as you have shown that there has been a clinically significant difference in the dyspnea scale between the two groups.

Following the Reviewer’s indication, we have added a comment at the beginning of the discussion section.

Reviewer 3 Report

Dear authors,

I read your manuscript with interest. HFNC has become an everyday device for the care of in-hospital patients while the extend of its applications have not been fully explored yet. There is quite an interest for the managment of patients with pulmonary hypertension at the moment.

Abstract: Well written.

Introduction: Well written.

Methods: Was the diagnosis of pulmonary hypertension already known or was a finding upon admission? Was the respiratory failure attributed to deterioration of PH? (in fig. 1 "acute on chronic respiratory failure associated with PH" so i suppose yes, but please add accordingly). 

Results: Were there under any other medications administered that could affect the result?

Discussion: Is the decrease of sPAP / changes in heart function considered clinically significant? Did the patients have better outcome?

Overall: Interesting and nocel topic. The manuscript need some improvement (pls see comments above).

Best regards.

Minor editing required.

Author Response

Dear authors,

I read your manuscript with interest. HFNC has become an everyday device for the care of in-hospital patients while the extend of its applications have not been fully explored yet. There is quite an interest for the managment of patients with pulmonary hypertension at the moment.

  • Abstract: Well written.

Thanks

  • Introduction: Well written.

Thanks

  • Methods: Was the diagnosis of pulmonary hypertension already known or was a finding upon admission? Was the respiratory failure attributed to deterioration of PH? (in fig. 1 "acute on chronic respiratory failure associated with PH" so i suppose yes, but please add accordingly).

As the Reviewer suppose, all patients were admitted for acute-on-chronic respiratory failure with a deterioration of an already known PH.

  • Results: Were there under any other medications administered that could affect the result?

No, there were not. Following also the indications of Reviewer #1, we have now specified that “Patients of the COT group were matched for age, sex, BMI, PH etiology and therapy.” (see Materials and methods).

  • Discussion: Is the decrease of sPAP / changes in heart function considered clinically significant? Did the patients have better outcome?

The study was unfortunately not designed to assess differences in clinical outcomes and the sample size is too small to demonstrate such differences. This is now discussed as study limitation.

  • Overall: Interesting and nocel topic. The manuscript need some improvement (pls see comments above).

Following the Reviewer’s indications, we have accordingly modified the manuscript. We thank he/she for the valuable comments and we hope now the manuscript fulfills the criteria for publication.

Reviewer 4 Report

Really good explanation about the effects of HFNC in right heart, mainly in reduction of PSP that in my opinion only needs small changes

Abstract: correct. It doesn´t need any change

Introduction: correct. It doesn´t need any change

Material and Methods: Correct. I think it should be explained better why They was only included 34 patients of 436, I suppose because it is a retrospective study and only this 34 had both cardiac ultrasonograhy, but it should be better explained

Results: correct. It doesn´t need any change

Discussion: This study suggests not demonstrate as you write. I think the explanation about why the PSP is reduced is correct, but initially or at the end it should be useful a summarize about that for better understand.

Conclusion: correct. It doesn´t need any change

Table 1: Female line is not necessary, it is enough with one of them, male or female.

Table 2: correct

Table 3: correct

C

Author Response

Really good explanation about the effects of HFNC in right heart, mainly in reduction of PSP that in my opinion only needs small changes

  • Abstract: correct. It doesn´t need any change

Ok, thanks.

  • Introduction: correct. It doesn´t need any change

Ok, thanks.

  • Material and Methods: Correct. I think it should be explained better why They was only included 34 patients of 436, I suppose because it is a retrospective study and only this 34 had both cardiac ultrasonograhy, but it should be better explained.

Following the Reviewer’s concern (and indications of Reviewer #1), we have added these data in the beginning of the Results section.

  • Results: correct. It doesn´t need any change

Ok, thanks.

  • Discussion: This study suggests not demonstrate as you write. I think the explanation about why the PSP is reduced is correct, but initially or at the end it should be useful a summarize about that for better understand.

Following the Reviewer’s concern, we have modified the text (see Discussion section)

  • Conclusion: correct. It doesn´t need any change

Ok, thanks.

  • Table 1: Female line is not necessary, it is enough with one of them, male or female.

Done, thanks.

  • Table 2: correct

Ok, thanks.

  • Table 3: correct

Ok, thanks.